# Breaking the photoswitch speed limit

Grace C. Thaggard[1], Kyoung Chul Park[1], Jaewoong Lim [1],
Buddhima K. P. Maldeni Kankanamalage[1], Johanna Haimerl[1,2], Gina R. Wilson[1],
Margaret K. McBride[1], Kelly L. Forrester[1], Esther R. Adelson[1], Virginia S. Arnold[1],
Shehani T. Wetthasinghe[1], Vitaly A. Rassolov [1], Mark D. Smith [1], Daniil Sosnin[3],
Ivan Aprahamian [3], Manisha Karmakar[4], Sayan Kumar Bag[4], Arunabha Thakur[4],
Minjie Zhang[5], Ben Zhong Tang [5,6,7,8], Jorge A. Castaño[9], Manuel N. Chaur[9,10],
Michael M. Lerch [11], Roland A. Fischer [2], Joanna Aizenberg [12,13],
Rainer Herges [14], Jean-Marie Lehn[15] & Natalia B. Shustova [1] ✉

The forthcoming generation of materials, including artificial muscles, recyclable and healable systems, photochromic heterogeneous catalysts, or tailorable supercapacitors, relies on the fundamental concept of rapid switching between two or more discrete forms in the solid state. Herein, we report a breakthrough in the "speed limit" of photochromic molecules on the example of sterically-demanding spiropyran derivatives through their integration within solvent-free confined space, allowing for engineering of the photoresponsive moiety environment and tailoring their photoisomerization rates. The presented conceptual approach realized through construction of the spiropyran environment results in ~1000 times switching enhancement even in the solid state compared to its behavior in solution, setting a record in the field of photochromic compounds. Moreover, integration of two distinct photochromic moieties in the same framework provided access to a dynamic range of rates as well as complementary switching in the material's optical profile, uncovering a previously inaccessible pathway for interstate rapid photoisomerization.

Rapid switching between two (or more) discrete states in the solid state is a cornerstone for the technological development of devices based on stimuli-responsive materials. For example, the development of on-demand-activated drug delivery platforms, photochromic heterogeneous catalysts, molecular motors, recyclable and healable materials, artificial muscles, multilevel anticounterfeiting and information encryption systems, and tailorable supercapacitors fully relies on fast changes between distinct states occurring in the solid state.[1–11]

[1]Department of Chemistry and Biochemistry, University of South Carolina, Columbia, South Carolina 29208, USA. [2]Chair of Inorganic and Metal-Organic Chemistry, Department of Chemistry, Technical University of Munich, Lichtenbergstrasse 4, 85748 Garching, Germany. [3]Department of Chemistry, Dartmouth College, Hanover, NH 03755, USA. [4]Department of Chemistry, Jadavpur University, 700032 Kolkata, India. [5]Department of Chemistry, Hong Kong Branch of Chinese National Engineering Research Center for Tissue Restoration and Reconstruction, and Guangdong-Hong Kong-Macau Joint Laboratory of Optoelectronic and Magnetic Functional Materials, The Hong Kong University of Science and Technology, Clear Water Bay, Kowloon, Hong Kong 999077, China. [6]School of Science and Engineering, Shenzhen Institute of Aggregate Science and Technology, The Chinese University of Hong Kong Shenzhen, Guangdong 518172, China. [7]Center for Aggregation-Induced Emission, South China University of Technology, Guangzhou 510640, China. [8]AIE Institute, Guangzhou Development District, Huangpu, Guangzhou 510530, China. [9]Departamento de Química, Universidad del Valle, AA 25360 Cali, Colombia. [10]Centro de Excelencia en Neuvos Materiales (CENM), Universidad del Valle, AA 25360 Cali, Colombia. [11]Stratingh Institute for Chemistry, University of Groningen, 9747 AG Groningen, The Netherlands. [12]Department of Chemistry and Chemical Biology, Harvard University, Cambridge, MA 02138, USA. [13]John A. Paulson School of Engineering and Applied Sciences, Harvard University, Cambridge, MA 02138, USA. [14]Otto Diels Institute of Organic Chemistry, University of Kiel, 24118 Kiel, Germany. [15]Laboratoire de Chimie Supramoléculaire, Institut de Science et d'Ingénierie Supraméléculaires (ISIS), Université de Strasbourg, 67000 Strasbourg, France. ✉e-mail: shustova@sc.edu

It could be anticipated that the forthcoming breakthroughs in artificial muscles, supercapacitors, and especially optoelectronics would critically hinge on substantial improvements in existing switching rates of stimuli-responsive building blocks.[2,6,7,9,11] For example, the area of ultra-efficient and high-speed optoelectronics is a rapidly growing field, which includes systems for high-speed data processing and information transport, multilevel anticounterfeiting and information encryption, as well as photodiodes and detectors operating at different timescales.[12] These listed industrially relevant applications are contingent upon the achievement of a substantial enhancement in photoisomerization rates. However, even in solution, where the molecules exhibit more degrees of freedom in comparison with the solid state, the rate constants of such processes are typically varied from $10^{-5}$ to $10^{0}\,s^{-1}$ using light as an external stimulus.[13,14] By utilizing solvent polarity and viscosity as variables, the speed of the photoisomerization process can be enhanced, but typically within one order of magnitude.[15] In the solid state, switching between distinct states is usually further constrained compared to solutions due to close packing, π-π stacking, or hydrogen-bonding interactions, especially for molecules for which isomerization is accompanied with large structural transformations or formation of zwitterionic species (e.g., spiropyran).[16,17]

Herein, we report a breakthrough in the isomerization speed limit of photochromic molecules on the example of sterically-demanding spiropyran derivatives, achieved through the employment of a conceptually distinct strategy, allowing for not only precise control of photoswitch environment (e.g., solvent-free), but also environment tunability due to matrix modularity. Furthermore, this approach led to a drastic enhancement of photoswitch isomerization ability in the solid

state, addressing challenges associated with limited photoisomerization due to strong intermolecular interactions typically pronounced in the solid state and impeding development of stimuli-responsive materials. To the best of our knowledge, such a rapid response observed in the solid state, through integration of spiropyran derivatives into the confined solvent-free space of rigid frameworks, has no analogs for these classes of photoswitches either in solution or in the solid state. While slight enhancements in photoisomerization rate could be anticipated and achieved using organic solvent[18] as a variable or by grafting certain types of photochromic molecules to a specific substrate[14], the presented conceptually different approach realized through construction of the spiropyran environment results in ~1000 times switching enhancement even in the solid state compared to its behavior in solution, setting a record in the field of photochromic compounds (Fig. 1). Using spectroscopic analysis in combination with photophysical measurements and theoretical modeling, we shed light on the mechanism of possible rapid photoisomerization. Thus, strategic design of light-responsive materials has resulted in rate enhancement that surpasses the preconceived "speed limit" for the commonly used classes of photoswitches, including but not limited to spiropyran-, diarylethene-, and hydrazone-based photochromic derivatives. Furthermore, the developed approach allows for integration of more than one type of photochromic molecule within the same platform, providing access to the development of solid-state materials with a very broad dynamic range of photoisomerization rates. Thus, the described studies represent a pathway for promoting rapid multivariable switching in the solid state that could be applied in many areas of chemistry and materials science as well as a great number of aforementioned practical applications.

## Results and discussion

Comparative analysis of the photochromic responses was performed for compounds belonging to three distinct classes possessing different photoisomerization mechanisms: spiropyran, hydrazone, and diarylethene derivatives (Fig. 2). Photoisomerization kinetics for each class was tested under the exact same experimental conditions, and the environment of the photochromic molecules was varied from solution or the solid state to solvent-free/solvent-filled porous rigid matrices. The latter contain substantial voids engineered to accommodate structural rearrangements necessary for photoswitch isomerization. The photochromic compound selection was performed based on the previously reported photoisomerization kinetic data to serve as points for comparison (Table 1 and Supplementary Table 10).[19–23] Selection of the previously studied compounds with reported photophysics was an avenue to deconvolute the influence of sample preparation and instrumentation (i.e., solution concentration, purity, or photon flux of the light source) on the targeted photophysical performance, especially considering the importance of wavelength selection for light-responsive molecules.[19–23]

Spiropyran derivatives, selected as the first class of photochromic molecules, possess well-established and rapid photoisomerization in solution, accompanied by substantial changes in their molecular conformation, optical properties, and dipole moment.[16,24] They undergo isomerization between neutral (spiropyran, SP) and charge-separated (merocyanine, MC) forms through excited-state C−O bond cleavage. Formation of the *cis*-merocyanine isomer through C−O bond cleavage is followed by thermal relaxation and rotation to the *trans*-merocyanine isomer (Figs. 2 and 3).[16,24] Thus, spiropyran photoisomerization is accompanied with both charge separation and significant structural rearrangement that hinders its photoisomerization in the solid state (Fig. 3), and therefore, limits its availability for material development. In contrast, diarylethene derivatives rely on a 6π electrocyclic rearrangement, resulting in ring closure and extension of π-conjugation throughout the backbone (Fig. 2).[25] Notably, this transformation occurs between two neutral isomers with minimal

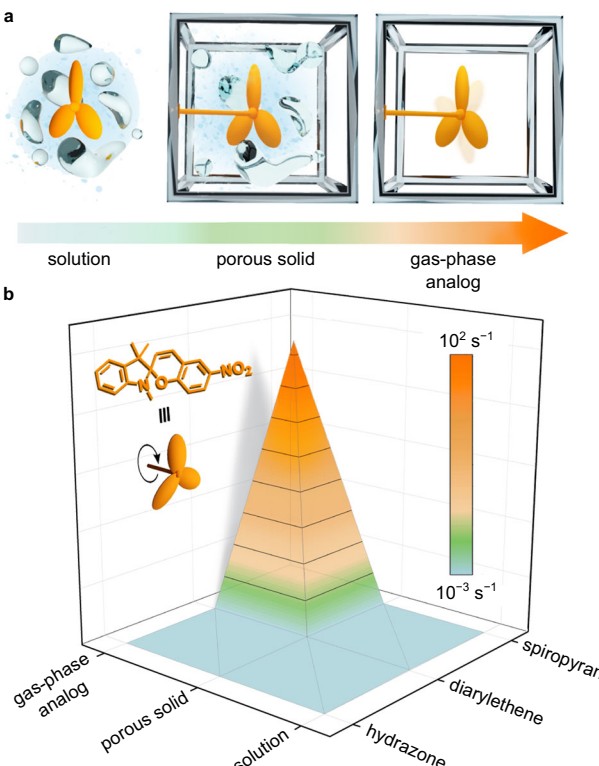

**Fig. 1 | Schematic representation of different photoswitch environments using a spiropyran derivative as an example. a** A spiropyran derivative (orange propellor) in (left) solution, (middle) a solvent-filled rigid solid matrix, and (right) a solid solvent-free porous matrix mimicking behavior in the gas phase (a gas-phase analog). **b** A plot demonstrating the drastic change in photoisomerization rate constants as a function of the photoswitch type and its environment. The exact values for photoisomerization rate constants are provided in Table 1.

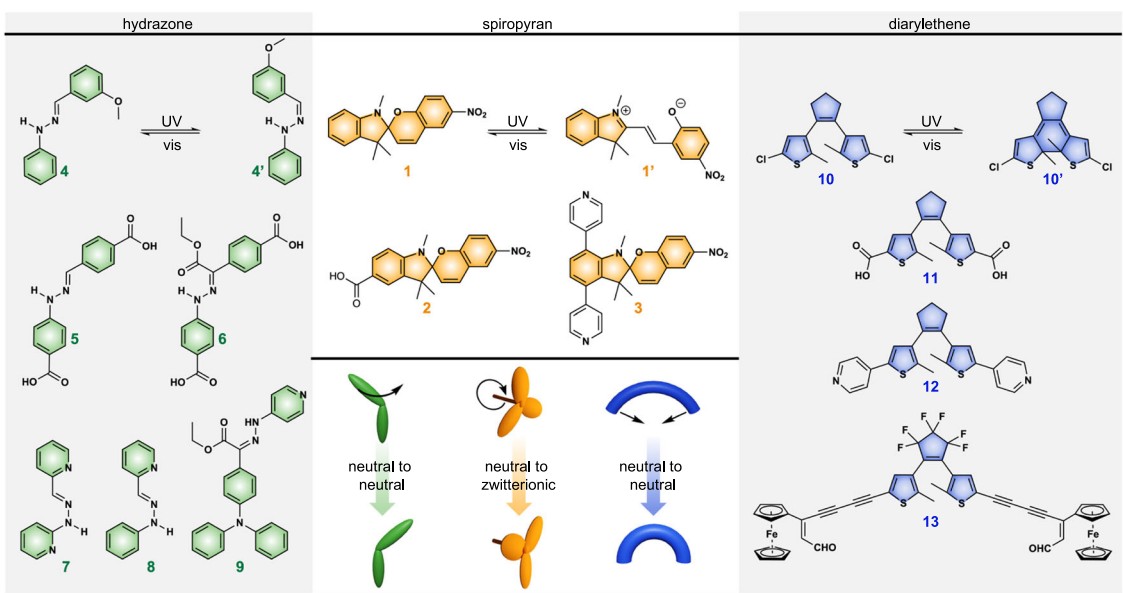

**Fig. 2 | Structures of spiropyran, hydrazone, and diarylethene derivatives used for photoisomerization kinetics studies.** Isomerization of spiropyran compounds (**1–3**, orange) is accompanied with significant structural rearrangement and zwitterionic merocyanine (**1'**) formation, whereas hydrazone (green) and diarylethene (blue) compounds undergo conversion between two neutral forms. Hydrazone compounds can undergo significant structural rearrangement upon photoisomerization while diarylethene derivatives do not.

**Table 1 | Summary of photoisomerization rate constants for spiropyran, hydrazone, and diarylethene derivatives in solution and coordinatively-integrated in frameworks**

| Sample | Conditions | $k$, s$^{-1}$ | Sample | Conditions | $k$, s$^{-1}$ |
|---|---|---|---|---|---|
| 1[a] | EtOH | 0.033(5) | 6[b] | DMF | 0.015(5) |
| 1[a] | toluene | 0.060(7) | UiO-67 + 6 | DMF | 0.029(3) |
| 1[a] | THF | 0.074(4) | UiO-67 + 6 | solvent-free | 0.026(2) |
| 1[a] | DMF | 0.14(2) | 7[b] | MeOH | 0.00060(7) |
| 2[a] | DMF | 0.036(2) | 8[b] | MeOH | 0.00060(8) |
| 2[a] | THF | 0.048(4) | 9[b] | toluene | 0.03(1) |
| UiO-67 + 2 | DMF | 0.061(4) | 10 | solid state | 0.13(3) |
| UiO-67 + 2 | solvent-free | 31.2(5) | 11 | solid state | 0.0337(5) |
| 3[a] | EtOH | 0.037(3) | 11[c] | DMF | 0.19(1) |
| 3[a] | toluene | 0.07(2) | UiO-67 + 11 | DMF | 0.32(2) |
| 3[a] | DMF | 0.08(1) | UiO-67 + 11 | solvent-free | 0.177(3) |
| 3[a] | CH$_3$CN | 0.054(4) | 12 | solid state | 0.36(6) |
| Zn$_2$(3)(DBTD) | DMF | 0.054(7) | 12[c] | DMF | 0.037(3) |
| Zn$_2$(3)(DBTD) | solvent-free | 53(2) | 12[c] | EtOH | 0.005(1) |
| 4[b] | toluene | 0.0086(3) | 12[c] | toluene | 0.009(1) |
| 4[b] | DMF | 0.0074(3) | Zn$_2$(12)(DBTD) | DMF | 0.0033(3) |
| 5[b] | DMF | 0.00484(1) | 13[c] | CH$_3$CN | 0.010(6) |
| 5[b] | EtOH | 0.016(3) | UiO-67 + 2 + 5 | DMF | 0.084(3)[d]0.0004(1)[e] |
| UiO-67 + 5 | DMF | 0.005(2) | UiO-67 + 2 + 5 | solvent-free | 55(2)[d] 0.00097(3)[e] |
| UiO-67 + 5 | solvent-free | 0.013(4) | | | |

[a]concentration for spiropyran derivatives in solutions was 3 mM;
[b]concentration for hydrazone derivatives 4, 5, and 6 in solutions was 30 μM and 7, 8, and 9 in solutions was 10 μM;
[c]concentration for diarylethene derivatives in solutions was 3 mM;
[d]photoisomerization rate constant corresponding to 2;
[e]photoisomerization rate constant corresponding to 5. For all samples, concentrations were selected based on the compound solubility and instrument set up. Standard deviations are provided in parentheses.

structural rearrangement. As a result, diarylethene derivatives frequently undergo efficient photoisomerization even in the solid state.[19] Hydrazone derivatives, as a third class of studied photochromic molecules, undergo photoisomerization through rotation or inversion around the C = N bond in the excited state (Figs. 2 and 3).[26,27] Similarly

to diarylethene compounds, hydrazone derivatives undergo isomerization between two neutral forms. However, isomerization of hydrazone derivatives is sometimes accompanied by significant structural rearrangements (Fig. 3) which are typically hindered by close packing or intermolecular forces in the solid state.[28,29] In such

significant structural rearrangement upon photoisomerization

**Fig. 3 | X-ray crystal structures of spiropyran and hydrazone derivatives. a** The SP (orthogonal geometry, CCDC 2283039) and MC (planar) forms of 2 demonstrate the significant structural rearrangement required for SP-to-MC photoisomerization which is accompanied with a -2.9 Å length increase. **b** Optimized structures of a hydrazone core, and the *E*-to-*Z* transformation is spatially demanding (approximated as a cone with a diameter of 6.4 Å for visualization)[17].

cases their applicability for fabrication of solid-state devices can be constrained.[30]

Photoisomerization kinetics experiments for the selected photochromic molecules shown in Fig. 2 were initially carried out in solution as a reference point for comparison with the literature reports (synthesis and corresponding references can be found in the Supplementary Note 1, Supplementary Methods, and Supplementary Table 10). All kinetics experiments were conducted at least in triplicate to confirm the validity of the obtained rate constants. Table 1 shows that the evaluated rate constants for photoisomerization processes in solution for the studied classes of photoswitches are in the range of 0.0006–0.2 s⁻¹, supporting the previously reported values.[19–23]

To promote isomerization of the sterically-demanding spiropyran and hydrazone derivatives in the solid state, we selected well-defined scaffolds (metal-organic frameworks (MOFs))[31–56] with 12–16 Å voids which can accommodate the significant structural changes (e.g., -90°-transformation of spiropyran, -180°-transformation of hydrazone, and length contraction/expansion of diarylethene derivatives; Fig. 3 and Supplementary Fig. 64) associated with isomerization. Three main criteria were employed for scaffold selection: (i) maintenance of scaffold integrity upon chemical modification and UV or visible irradiation, (ii) existence of voids suitable for unconstrained photoswitch isomerization, and (iii) presence of metal nodes allowing photoswitch coordination. Photoresponsive spiropyran and diarylethene derivatives with pyridyl and carboxylate groups, as well as hydrazone derivatives with carboxylate groups (e.g., spiropyran (2, 3) hydrazone (5, 6), and diarylethene derivatives (11, 12); Fig. 2) have been selected for incorporation through their coordination to the MOF unsaturated metal sites. Incorporation of stimuli-responsive units through metal-photoswitch binding prevents uncontrolled photoswitch leaching out of the material's pores. Indeed, our studies demonstrated that no leaching of photochromic molecules was detected according to spectroscopic analysis of the supernatant collected after submersion of the prepared photochromic MOFs into various organic solvents.

We have utilized two different synthetic approaches for photoswitch integration into the mentioned scaffolds: a de novo method that relies on integration of photoresponsive materials upon framework formation[19,50] and post-synthetic modification[17,57] that includes integration of targeted molecules after the scaffold synthesis. In the de novo approach, photoresponsive moieties were incorporated as pillars between two-dimensional layers constructed from zinc paddlewheel nodes connected by tetratopic H₄DBTD linkers (H₄DBTD = 3′,6′-dibromo-4′,5′-bis(4-carboxyphenyl)-[1,1′,2′,1″-terphenyl]−4,4″-dicarboxylic acid, Fig. 4). All synthetic procedures and related details are provided in the Supplementary Methods. In the second approach, carboxylic acid-functionalized photoresponsive linkers (e.g., spiropyran derivative 2, hydrazone derivatives 5 and 6, and diarylethene derivative 11) were post-synthetically coordinated to the unsaturated metal sites of a zirconium-based framework, $Zr_6O_4(OH)_4(BPDC)_6$ (BPDC²⁻ = 4,4′-biphenyldicarboxylate; UiO-67, UiO = University of Oslo), which possesses "defects", (i.e., missing ligands, Supplementary Figs. 12 and 63) that can be used for installation of carboxylic-containing linkers.[17,58,59] The integrity of all examined scaffolds upon photoswitch installation and/or irradiation was confirmed by powder X-ray diffraction analysis (PXRD, Supplementary Figs. 18–22). Photoswitch integration was evaluated by ¹H nuclear magnetic resonance (NMR) spectroscopy performed on the digested (destroyed in the presence of acid) MOF samples (Supplementary Figs. 13–17).

In the case of spiropyran derivatives, we used both synthetic methodologies (vide supra). Preparation of Zn₂(3)(DBTD) was carried out using the de novo approach (see Supplementary Note 1 for more synthetic details). This framework, consisting of two-dimensional DBTD⁴⁻ layers connected through $Zn_2(O_2C-)_4$ nodes, enables photoisomerization of sterically-demanding 3 by installation of 3 as pillars with photochromic moieties extending into the MOF pores (Fig. 4).[19,60] The second approach was performed through post-synthetic installation of 2 within the UiO-67 matrix, leading to formation of UiO-67 + 2. In both the de novo and post-synthetic modification approaches, the PXRD pattern of experimental Zn₂(3)(DBTD) matched the pattern simulated based on single-crystal X-ray diffraction data (Supplementary Fig. 23). The PXRD pattern of UiO-67 + 2 matched that of the parent UiO-67 framework as-synthesized as well as the simulated pattern, showing that the integrity of the scaffold was maintained after photoresponsive linker installation (Supplementary Fig. 18). ¹H NMR spectroscopic studies in combination with thermogravimetric analysis (TGA) confirmed integration of 2 within UiO-67, resulting in a framework containing 0.1 photoswitches per 10 metal nodes (Supplementary Fig. 13).

Following framework characterization, we studied photoisomerization kinetics of the prepared photochromic materials by fitting the change in absorbance as a function of time with a first-order exponential equation chosen based on previous literature reports.[19] For instance, the MC-to-SP conversion can be classically treated with a monoexponential decay function as shown before.[61] We compared the behavior for photoswitches integrated within the framework with the results obtained for the same photoswitches in solution. Notably, all solid-state samples (i.e., photochromic MOFs) were prepared such that the thickness of the samples did not exceed the penetration depth of the excitation wavelength. We analyzed the photoisomerization rates in the same solvents which were used for the framework synthesis, i.e., a photoswitch in solution and inside the MOF pores was surrounded by the same solvent molecules. Table 1 shows that the rate constants obtained for UiO-67 + 2 ($6.1 \times 10^{-2}$ s⁻¹) and Zn₂(3)(DBTD) ($5.4 \times 10^{-2}$ s⁻¹) are similar to the rates of 2 ($3.6 \times 10^{-2}$ s⁻¹) and 3 ($8.0 \times 10^{-2}$ s⁻¹) observed in *N,N*-dimethylformamide (DMF). Both spiropyran-based photoswitches integrated within the porous scaffolds exhibit reversible photoisomerization which also coincides with their behavior detected in solution, demonstrating that solution-like photoisomerization can be maintained in bulk solid materials. As a next step, we evaluated the possibility to modulate the photoisomerization kinetics of spiropyran derivatives through changes in solvent viscosity and polarity. Based on literature precedent, we anticipated that the presence of a polar solvent would stabilize the zwitterionic merocyanine isomer and impede the MC-to-SP isomerization process.[62,63] For instance, aqueous

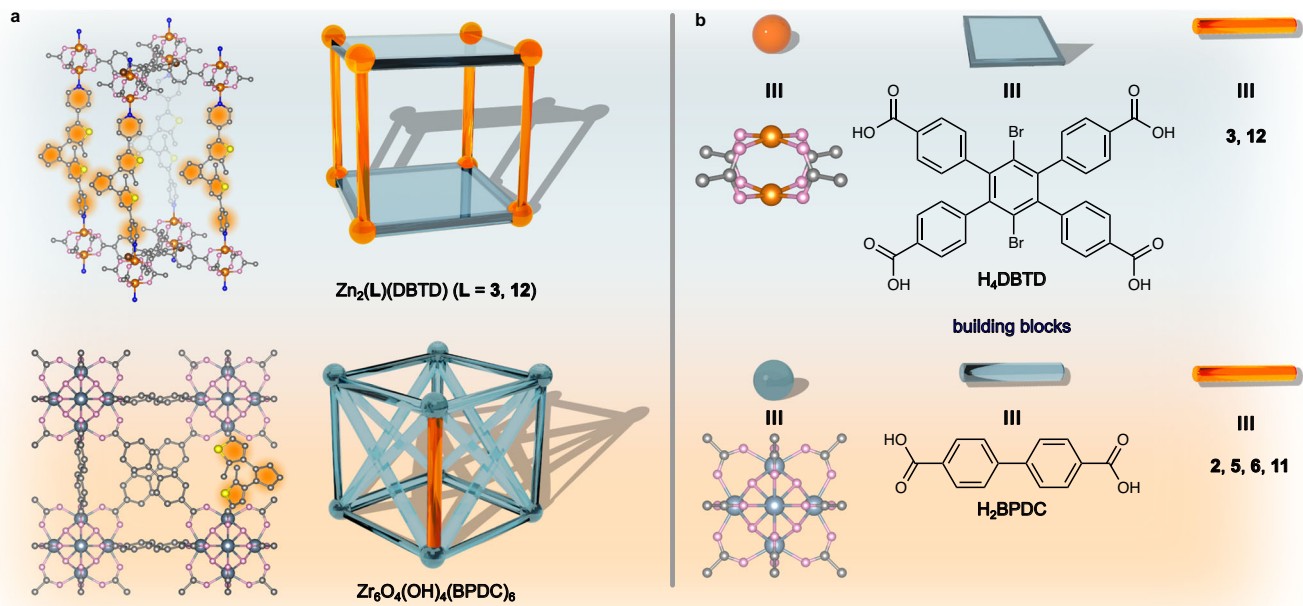

**Fig. 4 | Modes of photoswitch connectivity within MOF matrices. a** (top) X-ray crystal structure of Zn₂(12)(DBTD) and a 3D photochromic MOF structure consisting of 2D layers of H₄DBTD linkers (gray rectangle) connected by zinc paddlewheel metal nodes (orange spheres). Photoswitches (orange rod) are integrated as pillars between 2D layers. (bottom) X-ray crystal structure of Zr₆O₄(OH)₄(BPDC)₆ (UiO-67) with a possible mode of photoswitch incorporation highlighted in orange and a 3D photochromic UiO-67 MOF in which photoresponsive linkers (orange) are integrated through coordination to unsaturated zirconium nodes (see Supplementary Fig. 63 for additional views of the UiO-67 structure). **b** Building blocks for the preparation of the studied photochromic MOFs.

solutions of spiropyran derivatives are known to exist primarily in the merocyanine form because of the stabilizing hydrogen-bonding interactions.[64] Likewise, the presence of viscous solvent can inhibit the structural rearrangement upon spiropyran isomerization.[65] Indeed, we observed slight changes in the rate constant of MC-to-SP conversion depending on solvent choice (Table 1).

Since we observed that the solvent (e.g., 1 in ethanol compared to DMF) can impede the photoisomerization process by ~4-fold (Table 1), we evacuated the framework to "empty" the voids to advance the photoswitch isomerization rate. Notably, the photophysical measurements performed on solvent-free frameworks were conducted using identical experimental conditions as the measurements performed in solution or on solvent-filled frameworks, including excitation wavelength and photon flux. Indeed, the measurements performed on the evacuated scaffolds 2 and 3 revealed remarkable changes in photoisomerization rates (from $0.036 \pm 0.002$ s$^{-1}$ (2 in DMF) to $31.2 \pm 0.5$ s$^{-1}$ (UiO-67 + 2)), i.e., led to ~1,000-fold enhancement in comparison with the values measured in solution as demonstrated in Fig. 5 and Table 1. To the best of our knowledge, these values set a record for the photoisomerization rate enhancement of any spiropyran-based photoswitches in solution or in the solid state, providing a unique approach to engineer the environment that can lead to breakthroughs in the response acquired in stimuli-responsive solid-state materials. All the previous attempts to modulate the rate using the solvent viscosity or polarity led to very limited variations in rates according to the literature reports[64,66,67] or our studies performed for the exact same photoresponsive molecules, 2 and 3 (vide supra). A comparison of the found rates of related compounds reported in the literature and the rates obtained herein is reported in Supplementary Table 10.

One of the possible explanations for the observed drastic enhancement of MC-to-SP photoisomerization rates for 2 and 3 upon scaffold evacuation may be associated with the stabilization effect observed for the zwitterionic MC form in the presence of solvent molecules which could significantly impede the photoisomerization process.[15,67] To evaluate this hypothesis, we compared the obtained results for spiropyran compounds with hydrazone and diarylethene

derivatives. Isomerization of the latter ones is not associated with formation of zwitterionic species. As a first approach, we selected hydrazone derivatives, 5 and 6, for incorporation within the same rigid scaffold (UiO-67). Similar to 2 and 3, the isomerization of 5 and 6 requires significant structural transformations, however, it does not occur through formation of zwitterionic species (neutral *E*-hydrazone to neutral *Z*-hydrazone, Figs. 2 and 3). As a result, we expected to observe alterations in kinetics upon solvent removal caused primarily by steric factors rather than electronic effects, thereby mimicking the impact of viscosity. Taking into consideration steric requirements and the large rotation radius (~6.4 Å, Fig. 3) for hydrazone isomerization, i.e., significant void requirements necessary for installation of hydrazone derivatives, 5 and 6 were integrated within a UiO-67 matrix through the coordination of their carboxylate groups similar to 2 and 11 (vide infra). After installation, the material underwent a thorough washing procedure to remove the residual non-coordinated linker. The prepared UiO-67 + 5 and UiO-67 + 6 samples were then characterized by PXRD to ensure maintenance of framework integrity, and spectroscopic analysis (including ¹H NMR spectroscopy of the digested samples) was used to verify and quantify photoswitch installation (Supplementary Figs. 14 and 15).

To elaborate further, we also probed the kinetics of the coordinatively-integrated diarylethene compounds, 11 and 12, for which isomerization does not involve formation of charged species (similarly to 5 and 6 hydrazone derivatives) and occurs within the plane, i.e., without even significant structural rearrangements associated with isomerization of hydrazone or spiropyran derivatives. For instance, X-ray crystal structures of two photoisomers, demonstrating the minimal structural rearrangement associated with diarylethene isomerization, are presented in Supplementary Fig. 64. In the latter scenario, we expected to observe either no changes or minimal alterations in the photoisomerization kinetics upon MOF evacuation. To test our hypothesis, the pyridyl- and carboxylic acid-functionalized diarylethene derivatives were installed within the MOF using de novo and post-synthetic modification approaches, respectively (vide supra). For instance, 12 was incorporated as pillars between the two-

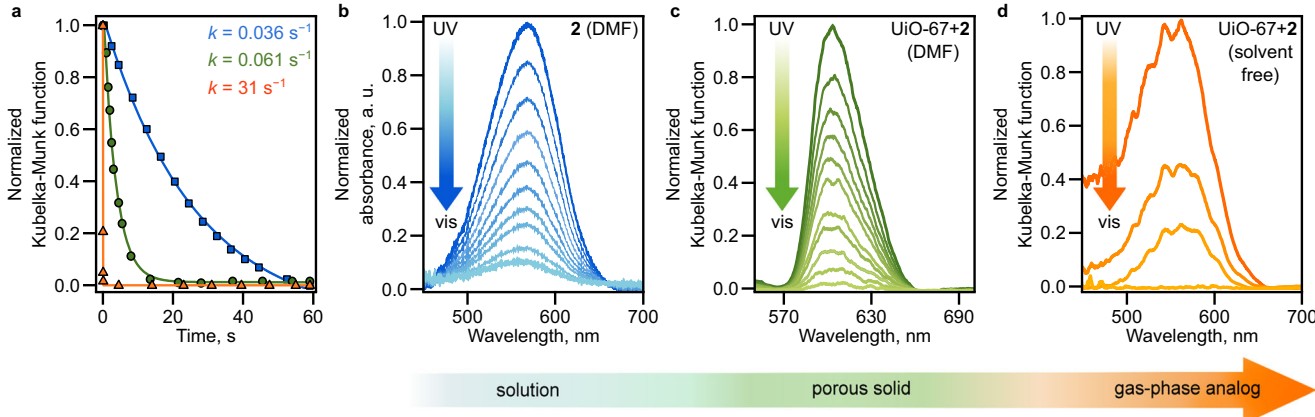

**Fig. 5 | Photoisomerization studies of a spiropyran derivative.**
**a** Photoisomerization kinetics studies of 2: in solution (blue), integrated in UiO-67 (green) containing DMF in the pores, and integrated in evacuated UiO-67 (solvent-free, orange). **b–d** Absorbance (blue) and diffuse reflectance (green, orange) spectra for 2 in each photoswitch environment. Fittings and corresponding rate constants are given in blue, green, and orange. The diffuse reflectance spectra for 2 in solution (**b**) and upon integration into UiO-67 (**c**) were collected at 500-ms intervals, while for evacuated UiO-67 (**d**), the suitable intervals were found to be 50 ms. In each measurement, the samples were irradiated with $\lambda_{ex} = 365$ nm for 30 s followed by attenuation under visible light ($\lambda_{ex} = 400$–900 nm).

dimensional layers formed from zinc paddlewheel metal nodes and planar $H_4$DBTD linkers in a similar fashion to 3, while 11 was anchored to the defect sites of UiO-67 (Fig. 4). However, evacuation of $Zn_2(12)$ (DBTD) resulted in framework degradation; therefore, this material was not suitable for studies of photoisomerization kinetics in a solvent-free environment.

As described above for the spiropyran derivatives, photophysical studies of hydrazone and diarylethene compounds integrated within the porous matrices revealed that each photoswitch maintained reversible photoisomerization with a rate constant comparable to the value obtained in solution. The rate constants determined in these experiments are summarized in Table 1. As anticipated, evacuation of UiO-67 + 5, UiO-67 + 6, and UiO-67 + 11 did not result in any significant rate enhancement in contrast to UiO-67 + 2 or $Zn_2(3)$(DBTD) containing spiropyran derivatives. Upon evacuation of the hydrazone-integrated MOFs, UiO-67 + 5 and UiO-67 + 6, we observed only a slight enhancement in the photoisomerization rate (~1.5-fold) in comparison with the rates found for the same photochromic compounds in DMF (Table 1), which we attributed to reduced solvent interactions analogous to a low-viscosity environment.[67] This rate increase is in line with the hypothesis that the structural rearrangement associated with hydrazone isomerization would be impacted by steric interactions with solvent molecules within the MOF pores, and therefore, the absence of solvent molecules within the MOF pores results in the rate increase. These findings are also in line with our assumption that the observed drastic differences in the rates of spiropyran derivatives (e.g., 2 and 3) are most likely associated with the transition from the zwitterionic merocyanine to neutral spiropyran form, allowing for the suppression of the solvent stabilization effect upon evacuation.

Similar to the discussed hydrazone compounds, diarylethene derivatives undergo photoisomerization between two neutral forms (Fig. 2). In addition, their isomerization occurs within the plane, and therefore, it is not limited by spatial constraints. As a result, we observed minimal variations in the measured rate constants in comparison with the solution values. These findings are also in line with our hypothesis that the rate enhancement detected for spiropyran derivatives is a result of both unhindered structural rearrangement and a lack of electrostatic solvent interactions stabilizing the merocyanine form. Using theoretical modeling, we also tested another parameter that could potentially result in the observed photoisomerization rate enhancement, in particular, the changes of the excitation energy barrier profile upon integration of spiropyran derivatives within a rigid matrix. Thus, we probed whether integration of the photochromic

molecules 2 and 3 in a rigid MOF matrix could alter the excited-state energy barrier for photoisomerization and associated kinetics. For this, we employed time-dependent density functional theory (TD-DFT) to construct and evaluate the excited-state potential energy surfaces (PESs) for 2 and 3, mimicking their behavior in solution and within a rigid framework (Fig. 6 and Supplementary Fig. 6). X-ray crystal structures of 2 and 3 serve as the initial points for full geometry optimization in the ground state. Then, the excited-state PESs were obtained using constrained geometry optimization in the relevant excited state while recording the changes in the $C_1$–O and adjacent $C_2$–O bond lengths ($r_1$ and $r_2$, respectively; Fig. 6 and Supplementary Figs. 3 and 4), which were used as the reaction coordinates (Fig. 6). The PESs were constructed by specifying the values of $r_1$ and $r_2$, then choosing and adjusting the coordinates of all unconstrained atoms to obtain the lowest energy value for the designated electronically-excited state using TD-DFT. During this optimization, $r_1$ and $r_2$ were fixed, and the outcome was a single point on the PES (see more details in Supplementary Note 2). This procedure was then repeated for different values of $r_1$ and $r_2$ in the range of interest. To probe the behavior of spiropyran derivatives coordinatively-integrated within the rigid framework, the coordinates of the corresponding anchoring groups (e.g., –COOH in 2 and pyridine groups in 3) were fixed as shown in Supplementary Fig. 5. Analysis of the PESs for either coordinatively-integrated or non-integrated spiropyran derivatives (i.e., mimicking either MOF or solution environments) showed minimal changes in their PES profiles (further details can be found in Supplementary Note 2). For instance, the computed excitation energies for coordinatively-integrated and non-integrated 2 are 26.57 and 26.66 kcal/mol, respectively. Thus, the observed rate enhancement for spiropyran derivatives could not be solely attributed to the effect of a rigid matrix but rather can be considered as a function of the detailed engineering of the pore environment.

Finally, we probed incorporation of two types of photochromic molecules, hydrazone and spiropyran, within the same platform to engineer a material with a broad spectrum of rates.[68] For that, 2 and 5 were successfully integrated within the UiO-67 scaffold through post-synthetic modification as described above. PXRD analysis showed that the framework integrity was maintained throughout the post-synthetic modification procedure, and $^1H$ NMR spectroscopic analysis in combination with TGA confirmed integration of both photoresponsive units, resulting in preparation of UiO-67 containing 0.1 and 0.6 of 2 and 5, respectively, per metal node (Supplementary Figs. 12, 17, and 22). Using spectroscopic analysis, we probed the distribution of

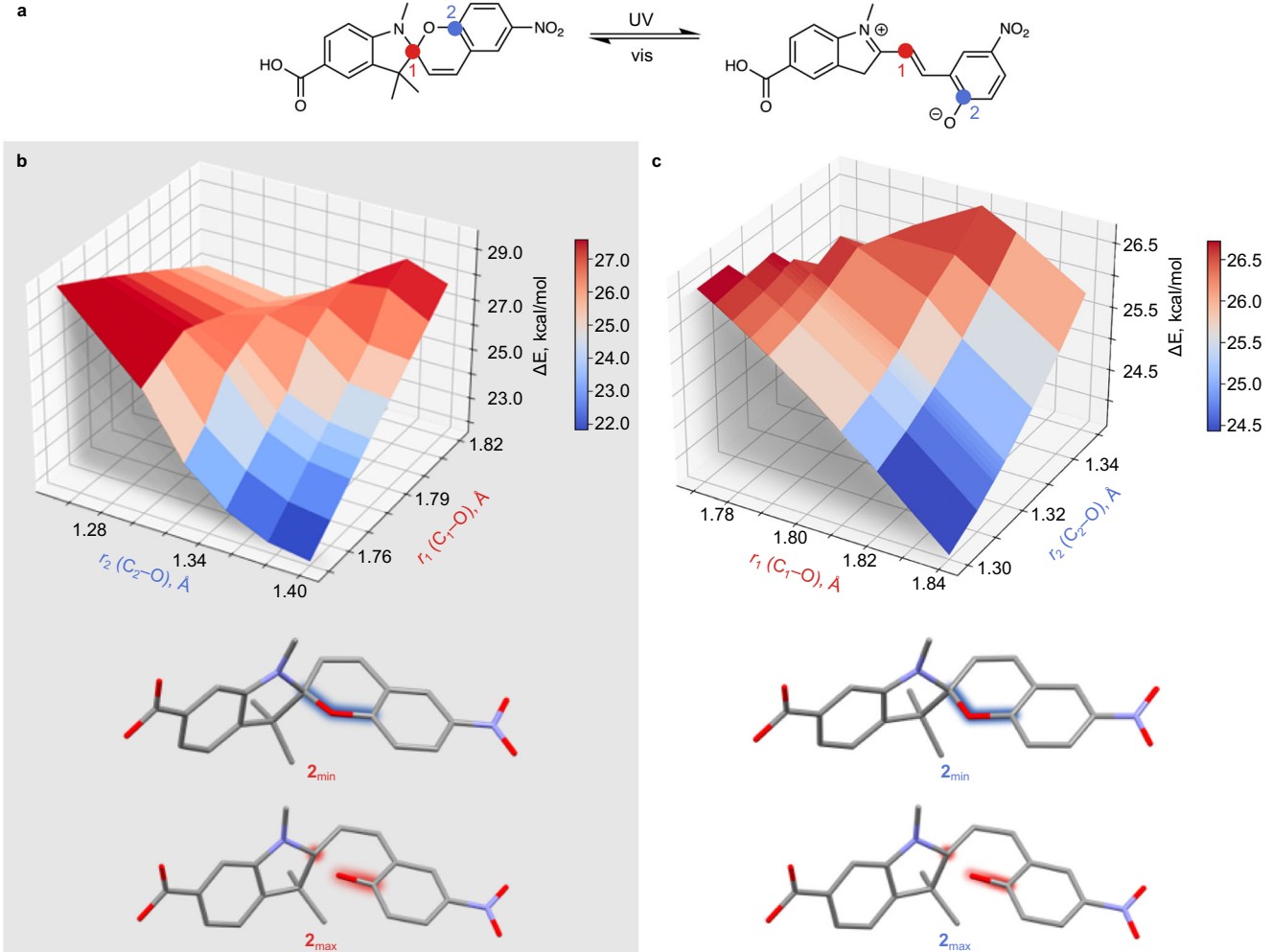

**Fig. 6 | Optimized structures of the spiropyran and merocyanine photoisomers of 2. a** The C1 and C2 atoms used for bond length monitoring are depicted as red (C1) and blue (C2) dots. **b** 3D PES for non-integrated 2 as a function of $r_1$ and $r_2$. **c** 3D PES for coordinatively-integrated 2 as a function of $r_1$ and $r_2$. The optimized geometrical conformations at the maximum and minimum energies for non-integrated and coordinatively-integrated 2 are highlighted in red and blue, respectively. Hydrogen atoms have been omitted for clarity. The maximum energy conformation for non-integrated 2 (**b**) occurs at $r_1 = 1.787$ Å, $r_2 = 1.318$ Å, and corresponding $\Delta E = 26.66$ kcal/mol. The maximum energy conformation for coordinatively-integrated 2 (**c**) occurs at $r_1 = 1.781$ Å, $r_2 = 1.318$ Å, and $\Delta E = 26.57$ kcal/mol.

photochromic molecules within the prepared material. We monitored emission of integrated 2 using epifluorescence microscopy, revealing that the prepared material is evenly fluorescent under 365-nm irradiation (Supplementary Fig. 60). In addition, we utilized energy dispersive X-ray spectroscopy (EDX) on the UiO-67 + 2 + 5 sample preexposed to a solution of $Cu(NO_3)_2$ in DMF under UV irradiation. It is known from our previous studies and literature reports[22,69,70] that the merocyanine photoisomer exhibits the ability to coordinate metal cations upon its exposure to solutions containing metal salts.[22,69,70] We used this approach for coordination of copper cations to the merocyanine form of 2. After a very thorough washing procedure to remove the residual copper cations, EDX spectroscopic analysis revealed a relatively homogeneous distribution of copper throughout the material (Supplementary Figs. 61 and 62). Thus, both spectroscopic techniques support a statistical distribution of spiropyran-based photoswitches installed in a rigid matrix. Since 2 and 5 were installed in the framework by the same mechanism and at the same time, we hypothesize that 5 is also statistically distributed throughout the material. As a next step, we evaluated the photophysical behavior of UiO-67 + 2 + 5 containing solvent molecules (Fig. 7) in the same way as we have performed for frameworks with only one type of photoresponsive fragment. Thus, the photoisomerization rate constant for 2

($\lambda_{max} = 576$ nm) was estimated to be $0.084 \pm 0.002$ s⁻¹ while the rate constant for 5 ($\lambda_{max} = 363$ nm) was estimated to be $0.0004 \pm 0.0001$ s⁻¹. Both of these values are in line with the photoisomerization rate constants obtained for the same photoresponsive moieties integrated within UiO-67 individually. Furthermore, incorporation of two classes of photochromic molecules in one material allowed for complementary modulation of distinct regions of the material's absorption profile. As anticipated, exposure to UV irradiation resulted in a simultaneous increase in absorbance in the visible region due to photoisomerization of 2 and decrease in absorbance in the UV region due to photoisomerization of 5. Thus, this material demonstrates a pathway for tuning the material's optical properties across a wide range of the UV-visible spectrum (300−650 nm, Fig. 7), allowing for control of the material's optical profile in a complimentary manner through simultaneous incorporation of distinct 2 and 5. Following photophysical characterization of as-synthesized UiO-67 + 2 + 5, we removed the solvent molecules from the framework pores through its evacuation for 48 h. As anticipated from the experiments with materials with only one type of photoresponsive unit, we observed a ~1000-fold increase in photoisomerization rate constant for the spiropyran derivative 2, while the rate of hydrazone 5 was increased slightly (~2.5-fold) as seen before for UiO-67 + 5 (Table 1

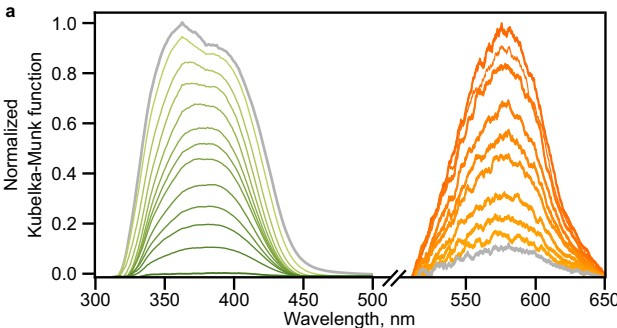
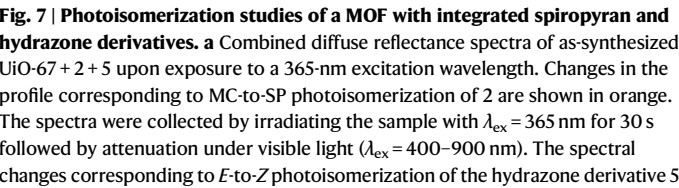
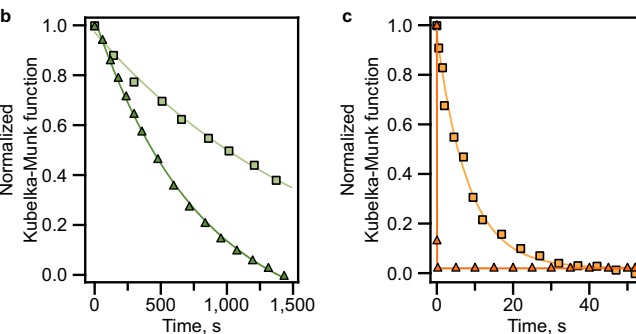

**Fig. 7 | Photoisomerization studies of a MOF with integrated spiropyran and hydrazone derivatives. a** Combined diffuse reflectance spectra of as-synthesized UiO-67 + 2 + 5 upon exposure to a 365-nm excitation wavelength. Changes in the profile corresponding to MC-to-SP photoisomerization of 2 are shown in orange. The spectra were collected by irradiating the sample with $\lambda_{ex}$ = 365 nm for 30 s followed by attenuation under visible light ($\lambda_{ex}$ = 400–900 nm). The spectral changes corresponding to $E$-to-$Z$ photoisomerization of the hydrazone derivative 5 are shown in green, and the spectra were collected under continuous UV irradiation ($\lambda_{ex}$ = 365 nm). **b** Kinetics studies of 5 photoisomerization in UiO-67 + 2 + 5 as-synthesized (squares) and after evacuation (triangles). Fittings are given in light and dark green, respectively. **c** Kinetics studies of 2 photoisomerization in UiO-67 + 2 + 5 as-synthesized (squares) and after evacuation (triangles). Fittings are given in light and dark orange, respectively.

and Fig. 7 and Supplementary Fig. 59). Thus, integration of two different classes of photoswitches into one scaffold provides access to a material with a wide dynamic range of rates in response to an external stimulus which, for instance, could offer a pathway to control release rates of multicomponent drugs in photopharmacology or can be employed for multilevel anticounterfeiting and information encryption.[10,11]

To summarize, we demonstrated a conceptually distinct approach to engineer photoswitch environment on the example of sterically-demanding spiropyran derivatives, resulting in ~1000-fold photoisomerization rate enhancement in the solid state, providing a pathway for the development of devices based on stimuli-responsive materials. Historically, achieving solution-like photoisomerization in a solid-state material has been a benchmark in the field of stimuli-responsive materials; however, the strategic engineering of the photoswitch environment through its integration in a rigid solvent-free matrix shown in this work is a pathway for surpassing solution-like behavior, leading to diminishment of stabilizing electrostatic interactions which are dominant and cannot be avoided in conventional solution-based studies. The presented approach also allows for integration of more than one type of photochromic unit within the same scaffold, preserving the achieved breakthrough in the photoisomerization rate of corresponding photochromic units. Moreover, incorporation of two distinct classes of photoresponsive units into the same host framework not only allows for selective tunability of one part of the absorption spectrum but also offers the opportunity to create a single material with a wide dynamic range of photoisomerization rates from $10^{-4}$–$10^{1}$ s$^{-1}$. The response time of such a material can be modulated over a range of $10^{5}$. Thus, the prepared framework, integrating distinct classes of stimuli-responsive units within the same platform, not only demonstrates a pathway for tuning and control the material's optical properties in a wide range (over 350 nm) but also provides a possible avenue to control, for instance, release rates of multicomponent drugs in photopharmacology[10] or multilevel anticounterfeiting and information encryption.[11] Therefore, the presented studies foreshadow a conceptually distinct avenue to achieve required breakthroughs in device performance, including development of probes for real-time monitoring of self-assembly processes[71] or biochemical pathways,[72] as well as near-instantaneous detection of, for instance, toxins.[73] However, these exciting applications only scratch the surface of the forthcoming generation of materials with tunable and tailorable properties that can be designed by leveraging the reported design principles and which can operate within a broad dynamic range.

## Methods

### Synthesis of 2,3,3-trimethyl-3*H*-indole-5-carboxylic acid (I, Supplementary Fig. 1)

The compound 2,3,3-trimethyl-3*H*-indole-5-carboxylic acid was prepared based on a modified literature procedure.[74] To a 250-mL round bottom flask, 3-methyl-2-butanone (2.20 mL, 20.5 mmol) and 4-hydrazineyl benzoic acid (2.00 g, 13.1 mmol), and acetic acid (25 mL) were added and heated at reflux for one day. After cooling to room temperature, the solvent was evaporated under reduced pressure. Then, 100 mL of acetone was added to the flask, and the reaction mixture was sonicated for 30 min. The filtrate was collected, and the solvent was removed under reduced pressure. Next, 100 mL of dichloromethane was added to the flask, and the resulting reaction mixture was sonicated for 30 min. The filtrate was collected, and the solvent was removed under reduced pressure, affording a red solid (1.82 g, 68%). The collected $^1$H NMR spectrum matched the reported one[74] (DMSO-$d_6$, 300 MHz): $\delta$ = 12.78 (s, 1H), 7.99 (s, 1H), 7.93–7.89 (d, 1H), 7.52–7.49 (d, 1H), 2.25 (s, 3H), and 1.27 (s, 6H) ppm.

### Synthesis of [5-carboxy-1,2,3,3-tetramethyl-3*H*-indol-1-ium] [trifluoromethanesulfonate] (II, Supplementary Fig. 1)

To a 250-mL round bottom flask, containing 100 mL of a mixture of diethyl ether and hexane (3:2 v/v), I (Supplementary Fig. 1, 1.00 g, 4.92 mmol), and methyl trifluoromethanesulfonate (1.37 mL, 12.5 mmol) were added. The resulting mixture was allowed to stir at room temperature for one day. Upon reaction completion, the precipitate was collected by vacuum filtration. The solid precipitate was transferred into a 250-mL round bottom flask, and then 100 mL of diethyl ether was added to the same flask. The resulting mixture was sonicated for 30 min to dissolve potential impurities. The precipitate was collected by vacuum filtration. The solid precipitate was transferred into a 250-mL round bottom flask, and 100 mL of isopropyl alcohol was added to the flask. The resulting mixture was sonicated for additional 30 min. The precipitate was collected by vacuum filtration, affording a solid (1.57 g, 87%). The collected $^1$H NMR spectrum matched the reported spectrum[74] (DMSO-$d_6$, 400 MHz): $\delta$ = 8.38 (s, 1H), 8.21–8.18 (d, 1H), 8.03–8.00 (d, 1H), 3.98 (s, 3H), 2.79 (s, 3H), and 1.56 (s, 6H) ppm.

### Synthesis of 1',3',3'-trimethyl-6-nitrospiro[chromene-2,2'-indoline]-5'-carboxylic acid (2)

To a 100-mL round bottom flask, II (Supplementary Fig. 1, 1.20 g, 3.27 mmol), 2-hydroxy-5-nitrobenzaldehyde (1.00 g, 5.98 mmol), pyridine (1.20 mL), and ethanol (15 mL) were added, and the reaction

mixture was refluxed for one day under nitrogen. After cooling to room temperature, 50 mL of diethyl ether was added to the solution, and the resulting mixture was stirred for one hour. The precipitate was collected via filtration and washed with hexane, affording a solid (0.980 g, 82%). [1]H NMR (DMSO-$d_6$): $\delta$ = 12.35 (s, 1H), 8.24 (s, 1H), 8.03–8.00 (d, 1H), 7.83–7.80 (d, 1H), 7.68 (s, 1H), 7.27–7.24 (d, 1H), 6.92–6.90 (d,1H), 6.70–6.68 (d,1H), 6.03–6.00 (d, 1H), 2.76 (s, 3H), 1.24 (s, 3H), and 1.13 (s, 3H) ppm. [13]C NMR (DMSO-$d_6$): $\delta$ = 167.37, 158.96, 151.20, 140.72, 135.88, 130.83, 128.50, 125.80, 122.94, 122.88, 121.60, 120.88, 118.78, 115.47, 106.27, 105.90, 51.56, 28.39, 25.48, and 19.53 ppm. Melting point: 285–287 °C. HR-ESI-MS: m/z found [M + H] for $C_{20}H_{19}N_2O_5^+$ 367.1290 (calculated 367.1294).

### Synthesis of 4-(2-(1-(4-carboxyphenyl)-2-ethoxy-2-oxoethylidene)hydrazineyl)benzoic acid (6)

To a 25-mL round bottom flask, 4-(2-ethoxy-2-oxoacetyl)benzoic acid (50.0 mg, 0.230 mmol) and 4-hydrazinobenzoic acid (41.0 mg, 0.270 mmol) were added and dissolved in 5.00 mL of ethanol. The resulting mixture was then stirred at room temperature for eight hours. After eight hours, the reaction mixture was diluted with 10.0 mL of water. The resulting precipitate was collected by filtration, and the solid was purified by column chromatography (0–50% ethyl acetate to hexanes), affording 6 as a yellow solid (49.0 mg, 62%). [1]H NMR (DMSO-$d_6$) $\delta$ = 11.77 (s, 1H), 7.97 (d, J = 8.1 Hz, 2H), 7.91 (d, J = 8.4 Hz, 2H), 7.79 (d, J = 8.1 Hz, 2H), 7.44 (d, J = 8.4 Hz, 2H), 4.40 (q, J = 7.1 Hz, 2H), 1.32 (t, J = 7.1 Hz, 3H). [13]C NMR (DMSO-$d_6$) $\delta$ = 167.53, 162.92, 147.45, 139.74, 131.65, 131.52, 130.59, 129.67, 127.95, 124.25, 114.07, 62.13, 14.31. Melting point: 272–273 °C. HR-ESI-MS: m/z found [M + H] for $C_{18}H_{17}N_2O_6^+$ 357.1079 (calculated 357.1087).

### Preparation of UiO-67 + 2

To postsynthetically integrate 2 into UiO-67, parent UiO-67 was first synthesized based on a literature procedure.[59] The UiO-67 powder was collected by filtration, thoroughly washed with DMF (3 × 10 mL), and then dried in air for 10 min. After drying, UiO-67 (25.0 mg, 11.8 μmol) was placed in a 20-mL vial. Then, 1.00 mL of a 30.0 mM DMF solution of 2 was added. The vial containing the resulting mixture was placed in a preheated aluminum block at 75 °C for 24 h. After cooling to room temperature, the resulting powder was collected by filtration and washed with DMF (3 × 10 mL). The PXRD patterns of parent UiO-67 and UiO-67 + 2 are shown in Supplementary Fig. 18. PXRD studies confirmed that MOF crystallinity was preserved after post-synthetic linker installation. The amount of 2 integrated in UiO-67 was calculated using a combination of TGA of the MOF and [1]H NMR spectroscopic analysis of the digested sample, which corresponds to 0.01 of 2 integrated per metal node. The [1]H NMR spectrum of the digested MOF sample confirming integration of 2 is shown in Supplementary Fig. 13.

### Preparation of UiO-67 + 2 + 5

To postsynthetically integrate both 2 and 5 in UiO-67, parent UiO-67 was first synthesized based on a literature procedure.[59] The UiO-67 powder was collected by filtration, thoroughly washed with DMF (3 × 10 mL), and then dried in air for 10 min. After drying, UiO-67 (25.0 mg, 11.8 μmol) was placed in a 20-mL vial. Then, 0.500 mL of a 30.0 mM DMF solution of 5 was added, and the mixture was heated at 75 °C for two hours. After two hours, 0.500 mL of a 30.0 mM DMF solution of 2 was added. The vial containing the resulting mixture was heated at 75 °C for 24 h. After cooling to room temperature, the resulting powder was collected by filtration and washed with DMF (3 × 10 mL). The PXRD patterns of parent UiO-67 and UiO-67 + 2 + 5 are shown in Supplementary Fig. 22. PXRD studies confirmed that MOF crystallinity was preserved after post-synthetic linker integration. The amount of 2 and 5 installed in UiO-67 was calculated using a combination of TGA of the MOF and [1]H NMR spectroscopic analysis of the digested sample, which corresponds to 0.1 of 2 and 0.6 of 5 integrated per metal node. The [1]H NMR spectrum of the digested MOF sample confirming integration of both 2 and 5 is shown in Supplementary Fig. 17.

### UV-vis absorbance spectroscopy

UV-vis absorbance spectra of hydrazone-containing samples in solution were collected using a ThermoFisher Evolution 350 UV-vis spectrometer with quartz cuvettes. Samples were irradiated with a mounted high-powered LED (M365L2, Thorlabs, $\lambda_{ex}$ = 365 nm, distance = 2.5 cm, and LEDD1B power supply set at 700 mA).

### Diffuse reflectance spectroscopy

Diffuse reflectance spectra for spiropyran- and diarylethene-containing samples were collected using an Ocean Optics JAZ spectrometer. An Ocean Optics ISP-REF integrating sphere was connected to the spectrometer using a 450-μm SMA fiber optic cable. For studies conducted under ambient atmosphere, the sample was loaded inside a 4.0-mm quartz sample holder. For studies conducted under dynamic vacuum, the sample was loaded into a quartz flow cell and connected to a Schlenk line with Tygon tubing. In all cases, a 400-nm longpass glass filter (Thorlabs, FGL400) was placed between the quartz sample holder and the integrating sphere to filter any UV light from the internal tungsten-halogen lamp. The quartz sample holder and longpass filter were attached to the top of the integrating sphere with electrical tape to prevent sample displacement. A mounted high-powered LED (M365L2, Thorlabs, $\lambda_{ex}$ = 365 nm, distance = 2.5 cm, and LEDD1B power supply set at 700 mA) was used for in situ irradiation of the samples. Diffuse reflectance spectra of hydrazone-containing samples were collected using a ThermoFisher Evolution 350 UV-vis spectrometer paired with Harrick Scientific Praying Mantis Diffuse Reflection accessory. For studies conducted under ambient atmosphere, the samples were prepared by filling the sample holder with $BaSO_4$ and placing 1–2 mg of the hydrazone-containing sample on top of the $BaSO_4$ layer. For studies conducted under vacuum, the samples were loaded into a high-temperature reaction chamber (Part HVC-VUV-5) and evacuated for 24 hours on a Schlenk line prior to collection of the spectra. A mounted high-powered LED (M365L2, Thorlabs, $\lambda_{ex}$ = 365 nm, distance = 2.5 cm, and LEDD1B power supply set at 700 mA) was used for in situ irradiation of the samples.

### Thermogravimetric analysis

TGA was used to determine the thermal stability of the synthesized UiO-67 and estimate the number of defects per metal node.[75] TGA was performed on an SDT Q600 thermogravimetric analyzer. Samples were loaded on an alumina boat as the sample holder at a heating rate of 5 °C/min to 600 °C under 10 mL/min air flow. A significant weight loss was observed during heating above 150 °C (Supplementary Fig. 12), which was used to calculate the number of defects per metal node based on a literature procedure.[74] Specifically, TGA was used to estimate the number of defects present in the UiO-67 samples by normalizing the weight loss to 100% at 600 °C using $ZrO_2$ as a reference. The "plateau" in the 300–400 °C temperature range was then used to estimate the number of linkers coordinated to the metal node.[74]

### Epifluorescence microscopy

Epifluorescence microscopic images of UiO-67 + 2 + 5 were collected using an Olympus BX51 microscope. Samples were irradiated for 30 s with a mounted high-powered LED (M365L2, Thorlabs, $\lambda_{ex}$ = 365 nm, distance = 2.5 cm, and LEDD1B power supply set at 700 mA). Images of the samples before and after UV irradiation (Supplementary Fig. 60) were collected with a color digital CMOS camera (Canon EOS REBEL T3/1100D).

## Scanning electron microscopy and energy dispersive X-ray spectroscopy (SEM-EDX)

To evaluate the distribution of photochromic molecules integrated into UiO-67, a 5.00-mg sample of UiO-67 + 2 + 5 was exposed to 2.00 mL of a solution of $Cu(NO_3)_2$ (0.60 mM in DMF) under UV light ($\lambda_{ex} = 365$ nm) for three days to promote coordination of copper cations by merocyanine.[22] After three days, the sample was washed thoroughly with DMF and ethanol. Following the washing procedure, the sample was dried on vacuum overnight. The sample was mounted on a carbon substrate and analyzed with a Tescan Vega 3 SBU variable pressure SEM equipped with a backscattered electron detector. EDX analysis was performed with an accelerating voltage of 20.0 kV, a beam intensity of 13, a working distance of 10 mm, and 1962× magnification. EDX analysis revealed an equal distribution of copper in the sample (Supplementary Fig. 61). The obtained value for copper cation integration is in line with the amount of installed spiropyran. Based on elemental mapping, the copper cations bound by merocyanine and coordinated to remaining defect sites were evenly distributed throughout the bulk material (Supplementary Fig. 62).

## Other physical measurements

NMR spectra were obtained on Bruker Avance III-HD 300, Bruker Avance III 400 MHz, Oxford 500 MHz, and Bruker Ascend 600 MHz NMR spectrometers. $^{13}C$ and $^1H$ NMR spectra were referenced to natural abundance $^{13}C$ peaks and residual $^1H$ peaks of deuterated solvents, respectively. Chemical shifts are quoted in ppm relative to tetramethylsilane (TMS), using the residual solvent peak as the reference standard. FTIR spectra were collected on a Perkin-Elmer Spectrum 100. PXRD patterns were recorded on a Rigaku Miniflex II diffractometer at a scan rate of 5°/min with accelerating voltage and current of 40 kV and 15 mA, respectively.

## Data availability

The main data supporting the findings of this study are available within the paper and its Supplementary Information. Source data are provided with this paper. The crystallographic data generated in this study have been deposited in the Cambridge Crystallographic Data Centre under accession code 2283039.

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

## Acknowledgements

The authors are grateful for support from the NSF Award (DMR-2103722) and SC EPSCoR GEAR. N.B.S. also acknowledges support from the Dreyfus Teaching-Scholar Award supported by the Dreyfus Foundation as well as the USC ASPIRE Award. G.C.T. is supported by the National Science Foundation Graduate Research Fellowship under Grant No. DGE-2034711. DMR-2103722, N.B.S. SC EPSCoR GEAR, N.B.S. Dreyfus Teaching-Scholar Award, N.B.S. USC ASPIRE Award, N.B.S. DGE-2034711, G.C.T.

## Author contributions

G.C.T. synthesized and characterized photochromic molecules and frameworks, performed photoisomerization kinetics studies, performed formal analysis of kinetics data, developed methodology, created illustrative materials, and wrote the original draft. K.C.P. synthesized and characterized UiO-67 frameworks and performed formal analysis of kinetics data. B.K.P.M.K. and G.R.W. characterized the photochromic frameworks. J.H. and R.A.F. created illustrative materials. M.K.M, K.L.F., E.R.A., and V.S.A. performed photo-isomerization kinetics studies in solution. S.T.W. and V.A.R. performed theoretical modeling. M.D.S. collected and solved the single-crystal X-ray structures. J.L, D.S., I.A., M.K., S.K.B., A.T., M.Z., B.Z.T., J.A.C., M.N.C., J.-M.L., M.M.L., and J.A. synthesized and characterized photochromic molecules. R.H. contributed to the scientific discussion and preparation of the manuscript. N.B.S. conceptualized the studies, acquired funding, wrote and edited the original draft, and administered the project.

## Competing interests

The authors declare no competing interests.
