## [Peer Review File · Nature Communications]

Breaking the Photoswitch Speed LimitREVIEWER COMMENTS

Reviewer #1 (Remarks to the Author):

Thaggard et al. present a comprehensive study on boasting advantages of frameworks to probe essential requirements for highly efficient photochromism. The authors presented their results which conclude the importance of void space to release physical constraints upon isomerization. The work chose frameworks smartly to investigate the steric and electronic effect, and the choice of the photochromic molecules well represents the field's interest while touching upon the critical difference to evaluate different types of variables towards the efficiency of photochromism. Thus, I think the work presents a novel systemic design to prove their hypothesis, wherein the results are remarkable for solving a challenging problem. Beyond that, the study provides fundamental insights into advancing the research community and the broad readership of Nature Communications. I recommend publishing this work after the following minor comments are adequately addressed.

1. In Figure 2, although the authors defined acronyms properly, it might help to follow easier if labels of SP and MC could be reintroduced in the figure or the caption.
2. For Figure 2, can the authors explain the delta in the molecular length regarding the photochromism?
3. In Figure 5, the authors may confirm the E/Z isomers of the top reaction scheme with structures presented in their calculations.
4. Lastly, it might be practically unfeasible, but I wonder if pore size distribution can capture the different states of photo isomer for the most representative sample to corroborate the confinement effect.

The manuscript is well-written and tackles an important problem with great experimental designs.

Reviewer #2 (Remarks to the Author):

In this research, the team investigates the impact of confining molecules within metal-organic frameworks (MOFs) and the effect of nearby solvents on solid-state photoswitching dynamics. A standout finding is the incredibly fast switching from merocyanine to spiropyran in empty MOFs, which operates approximately 1000 times faster than in a solution or MOFs containing solvents. It's believed that this amplified speed results from multiple elements, including the absence of solvent-induced stabilization for the zwitterionic merocyanine and the lack of a viscous solvent slowing molecular structural shifts. When applying the same conditions across various well-known photoswitches, such as hydrazones and diarylethenes, this rapid switching was only observed with spiropyran-merocyanine pairs in the solvent-free frameworks, attributed to their significant polarity shift during SP-MC switching. The research effectively illustrates this new principle with detailed optical and structural analyses. A major takeaway from this study is the evidence that solid-state switching speeds can surpass solution-state, a notion which challenges conventional understanding and points to the importance of precise spatial

control of photoswitches in nanoporous frameworks.

Before publishing this research, the authors should consider the following technical questions:

1. The arrows' direction between spiropyran and merocyanine in Figure 2 needs to be revised.
2. The depicted photoswitching between E and Z isomers of hydrazones in Figures 1 and 2 might be too general. The effect of specific chemical structures and functional groups, as seen in Figure 1, can impact this. Authors should consider certain exceptions, like compound 6 (Z is thermodynamically stable and responds to visible light), and provide more detail on the UV-irradiation experiments in the supplementary information (Figures S39-41).
3. The cartoon representation of UiO-67 in Figure 3 doesn't effectively depict the diagonally positioned BPDC ligands in the X-ray structure.
4. Figure 4's presentation of compound 2's absorbance change could be clearer. It was only after referring to the supplementary figure captions that the sequence of a short UV irradiation followed by visible irradiation became evident. This should be better explained in the main content.
5. Line 287 "the observed drastic differences in the rates of spiropyran derivatives (e.g., 2 and 3) are most likely associated with the transition from the neutral spiropyran to the zwitterionic merocyanine form, allowing for the suppression of the solvent stabilization effect upon evacuation." should be changed to "... transition from the zwitterionic merocyanine to the neutral spiropyran form..."
6. Figure 6 needs clearer labeling on the experimental conditions, especially regarding that the orange curves were obtained upon short 365 nm excitation followed by visible light irradiation and that the green curves were obtained under the continuous UV irradiation. If there are misunderstandings about this process, further clarification would be beneficial.
7. The UV-induced transition from SP to MC in the multi-photoswitch MOF experiment wasn't compared to its counterpart in solutions or solvent-containing MOFs. The authors' insights on this would be valuable.

Reviewer #3 (Remarks to the Author):

This report studied the dynamics of the photochromic molecules, including spiropyran (SP) to merocyanine (MC), by providing a new structure for the formation of the assembly and neighboring environment. As a background, the isomerization rate of the photochromic molecules is critical for their application to the switching material. Authors claim that the relatively slow switching time of SP-MC photoisomerization in solution is due to the interaction of the photochromic molecules with neighboring solvents, partially due to the structure change in the isomerization process between SP and MC. Thus, a

gaseous condition offers a small interaction with the neighbors and accelerates the isomerization process. The authors constructed an MOF structure whose pillars the SP molecules form. The concept has a novelty with an intriguing result for accelerating the switching time by a significant order. In addition, the material and property analysis details are well described in detail. This paper should attract wide attention and should be published. However, before publication, the authors should revise the following points.

1. Though the authors determine the conversion rate for all possible combinations of the photochromic molecules and the neighboring environment, no temperature dependence is examined to access the activation energy, which should be important data to compare with calculation and so on
2. The authors attribute the significant increase in the switching rate to the neighboring change from the solution to the vacuum. Since the authors are describing the energy difference of several states of the isomerization, the referee considers that the activation barrier difference with the change of the surroundings from the solution to the vacuum can be estimated with recently developed DFT calculations. The accuracy required is to discuss the energy difference's order and judge whether the rate increase is rational or not.
3. The following report compares the SP MC activation barrier in the solution and adsorbed on a solid surface. The authors should consider citing the article. "Mamun et al. Chemistry of the photoisomerization and thermal reset of nitro-spiropyran and merocyanine molecules on the channel of the MoS₂ field effect transistor. *Phys. Chem. Chem. Phys.* 2021, 23, (48), 27273-27281"
4. The paragraph starting from 248 has a length of a page. Though the authors describe in detail of the similarities and differences between molecules, the referee had a hard time understanding the flow of the story. This is also true for the following two paragraphs, which start with the same phrase of 'Similar to'. The authors should check whether the description can be more organized or not.

Reviewer #4 (Remarks to the Author):

The submitted manuscript (NCOMMS-23-33281) was reviewed and it could be accepted for publication in Nature Communications after considering the following major corrections:

1. At the beginning of "Introduction", Ref #1-11 has been cited twice. It is better to split them and specify each to the relevant subject.
2. Page 2, Line 51-52: Please note that what you are talking about is "rate constant" and not "rate".
3. "Scheme 1" has not been cited in the text. Also, the description for the right scheme in caption ((right) a solid solvent-free...) does not conform with the scheme (gas-phase analog).
4. Page 5, Line 113-115: It is suggested to provide crystallographic data for diarylethens to confirm the statement about their minimal structural rearrangement.
5. Page 7, Line 144-145: Please check that the hydrazones with pyridyl groups are 7-9 (Figure 1). Also, check Table 1 for this issue.
6. Table 1: What are the numbers in parenthesis in the "k" column, i.e. 0.033(5), 0.06(7) and etc.? Define them properly in the text or Table footnote.
7. Figure 4: For better elucidation and conformation with the text and Table 1, you may replace

“evacuation” (in the caption) or “(evac)” (in the right spectrum) with “solvent-free”.

8. Page 12, Line 255-260: The authors talk about steric factors for hydrazones as the controlling parameter on the kinetic of photoisomerization. Why didn't they note to the dipole-dipole interactions for E and Z isomers during such photo-transformation? This seems to be important for hydrazones, while they would be ignored for diarylethens. However, this parameter and more have comprehensively been studied before for spiropyran (J. Mater. Chem. C, 2017, 5, 6588—6600) and the authors may refer to it.

9. In continuum to comment #8 and with respect to Table 1, the authors have not assessed the role of variation in solvent polarity on the photoisomerization process. It is recommended to give rate constants for one of the hydrzones in solvents with different polarities. This may improve their discussion and deduction about the interaction of E and Z isomers with the employed MOF groups.

10. The authors need to explain the importance of drastic enhancement in the photoisomerization rate (constant) in “Introduction” and “Results and discussion” properly. What applications will be developed and weaknesses will be covered by this potentiality? Some few points have been mentioned in “Conclusion” very briefly, but I think it needs to be emphasized more.

The Reviewers provided excellent suggestions and corrections that we have carefully considered and addressed. Below are our replies to each point raised by the Reviewers.

Reviewer 1

“Thaggard et al. present a comprehensive study on boasting advantages of frameworks to probe essential requirements for highly efficient photochromism. The authors presented their results which conclude the importance of void space to release physical constraints upon isomerization. The work chose frameworks smartly to investigate the steric and electronic effect, and the choice of the photochromic molecules well represents the field’s interest while touching upon the critical difference to evaluate different types of variables towards the efficiency of photochromism. Thus, I think the work presents a novel systemic design to prove their hypothesis, wherein the results are remarkable for solving a challenging problem.”

We thank Reviewer 1 for their careful reading and appreciation of our work.

“Beyond that, the study provides fundamental insights into advancing the research community and the broad readership of Nature Communications. I recommend publishing this work after the following minor comments are adequately addressed.”

We appreciate Reviewer 1’s recommendation to publish our work and for the helpful comments which we address below.

“In Figure 2, although the authors defined acronyms properly, it might help to follow easier if labels of SP and MC could be reintroduced in the figure or the caption.”

We completely agree with Reviewer 1 and have reintroduced these labels in the caption of Figure 2.

“For Figure 2, can the authors explain the delta in the molecular length regarding the photochromism?”

Reviewer 2 makes an excellent point that the length and volume occupied by spiropyran derivatives change upon photoisomerization. For compound **2** (Figure 2), the distance between the carboxylic groups and the carbon adjacent to the nitro group increases from 10.6 to 13.5 Å upon photoisomerization from the spiropyran to merocyanine forms, respectively. We have highlighted this aspect in the revised version of the manuscript.

“In Figure 5, the authors may confirm the E/Z isomers of the top reaction scheme with structures presented in their calculations.”

We thank Reviewer 1 for pointing out this inconsistency. We have presented the optimized structures used for theoretical calculations in the revised version of Figure 5.

“Lastly, it might be practically unfeasible, but I wonder if pore size distribution can capture the different states of photo isomer for the most representative sample to corroborate the confinement effect.”

Reviewer 1 makes a very interesting point. As shown by Klajn and co-workers, equilibrium between spiropyran and merocyanine forms (and corresponding rate constants) of spiropyran derivatives could be tailored based on their inclusion within metal-organic cages with different pore geometries.^[R1] Thus, it is reasonable to suggest that altering the pore environment in MOFs could potentially affect the photoisomerization rate constants. However, it would be extremely difficult to prove the position (occupation of specific pores) by spiropyran due to significant crystallographic disorder. That is why demonstrating such effect utilizing discrete systems such as cages (similar to Klajn’s work) is more feasible. Studies of photochromic molecules integrated in cages with different pore sizes and shapes are currently underway in our laboratory.

“The manuscript is well-written and tackles an important problem with great experimental designs.”

We appreciate Reviewer 1’s kind comments about our work.

Reviewer 2

“In this research, the team investigates the impact of confining molecules within metal-organic frameworks (MOFs) and the effect of nearby solvents on solid-state photoswitching dynamics. A standout finding is the incredibly fast switching from merocyanine to spiropyran in empty MOFs, which operates approximately 1000 times faster than in a solution or MOFs containing solvents. It’s believed that this amplified speed results from multiple elements, including the absence of solvent-induced stabilization for the zwitterionic merocyanine and the lack of a viscous solvent slowing molecular structural shifts. When applying the same conditions across various well-known photoswitches, such as hydrazones and diarylethenes, this rapid switching was only observed with spiropyran-merocyanine pairs in the solvent-free frameworks, attributed to their significant polarity shift during SP-MC switching. The research effectively illustrates this new principle with detailed optical and structural analyses. A major takeaway from this study is the evidence that solid-state switching speeds can

surpass solution-state, a notion which challenges conventional understanding and points to the importance of precise spatial control of photoswitches in nanoporous frameworks.”

We thank Reviewer 2 for their careful reading of our work.

“The arrows' direction between spiropyran and merocyanine in Figure 2 needs to be revised.”

We thank Reviewer 2 for pointing out this oversight and have corrected the direction of the arrow in Figure 2.

“The depicted photoswitching between E and Z isomers of hydrazones in Figures 1 and 2 might be too general. The effect of specific chemical structures and functional groups, as seen in Figure 1, can impact this. Authors should consider certain exceptions, like compound 6 (Z is thermodynamically stable and responds to visible light), and provide more detail on the UV-irradiation experiments in the supplementary information (Figures S39-41).”

We completely agree with Reviewer 2 that it is misleading to generalize whether the *E*- or *Z*-isomer of a hydrazone derivative is formed by UV or visible irradiation due to the dependence on the specific chemical structure. For this reason, we have labeled the excitation wavelengths in these figures based on the photoisomers shown on either side of the arrows. We have included additional information about the individual hydrazone derivatives in the figure captions of the revised Supplementary Information (Supplementary Fig. 34–45).

“The cartoon representation of UiO-67 in Figure 3 doesn't effectively depict the diagonally positioned BPDC ligands in the X-ray structure.”

We thank Reviewer 2 for pointing out the inaccuracy of the schematic MOF representation. We have revised Figure 3 accordingly. In addition, we have included the X-ray crystal structure of UiO-67 in the Supplementary Information (Supplementary Fig. 61).

“Figure 4's presentation of compound 2's absorbance change could be clearer. It was only after referring to the supplementary figure captions that the sequence of a short UV irradiation followed by visible irradiation became evident. This should be better explained in the main content.”

We appreciate Reviewer 2's suggestion which will definitely improve the clarity of our manuscript. We included an explanation of how these spectra were collected in the revised caption of Figure 4.

“Line 287 “the observed drastic differences in the rates of spiropyran derivatives (e.g., 2 and 3) are most likely associated with the transition from the neutral spiropyran to the zwitterionic merocyanine form, allowing for the suppression of the solvent stabilization effect upon evacuation.” should be changed to “... transition from the zwitterionic merocyanine to the neutral spiropyran form...””

We completely agree with Reviewer 2 and have changed the wording of this sentence according to their suggestion.

“Figure 6 needs clearer labeling on the experimental conditions, especially regarding that the orange curves were obtained upon short 365 nm excitation followed by visible light irradiation and that the green curves were obtained under the continuous UV irradiation. If there are misunderstandings about this process, further clarification would be beneficial.”

We thank Reviewer 2 for pointing out ways to improve the understanding of our studies. We have included a brief description of the experimental conditions for the mentioned measurements in the revised caption of Figure 6.

“The UV-induced transition from SP to MC in the multi-photoswitch MOF experiment wasn't compared to its counterpart in solutions or solvent-containing MOFs. The authors' insights on this would be valuable.”

We appreciate Reviewer 2's point and interest in this subject, however the reported rate constants for all spiropyran derivatives correspond to the reverse (i.e., merocyanine to spiropyran) process. We clarified this aspect in the revised version of the manuscript.

Reviewer 3

“This report studied the dynamics of the photochromic molecules, including spiropyran (SP) to merocyanine (MC), by providing a new structure for the formation of the assembly and neighboring environment. As a background, the isomerization rate of the photochromic molecules is critical for their application to the switching material. Authors claim that the relatively slow switching time of SP-MC photoisomerization in solution is due to the interaction of the photochromic molecules with neighboring solvents, partially due to the structure change in the isomerization process between SP and MC. Thus, a gaseous condition offers a small interaction with the neighbors and accelerates the isomerization process. The authors constructed an MOF structure whose pillars the SP molecules form. The concept has a novelty with an intriguing result for accelerating the switching time by a significant order. In addition, the material and property analysis details are well described in detail. This paper should attract wide attention and should be published.”

We really thank Reviewer 3 for thorough reading of our studies and appreciation of our work.

“Though the authors determine the conversion rate for all possible combinations of the photochromic molecules and the neighboring environment, no temperature dependence is examined to access the activation energy, which should be important data to compare with calculation and so on.”

We completely agree with Reviewer 3 that VT measurements would provide valuable insight. However, the low activation barrier of spiropyran photoisomerization in the solvent-free environment even at room temperature was measured at the limit of our optical spectrophotometer and overall set up. We are building a new instrument to explore this opportunity at elevated temperatures to study property-temperature correlation. In addition, we anticipate that these studies would be very lengthy (at least the length of the current manuscript), and we plan them to include into a separate manuscript.

“The authors attribute the significant increase in the switching rate to the neighboring change from the solution to the vacuum. Since the authors are describing the energy difference of several states of the isomerization, the referee considers that the activation barrier difference with the change of the surroundings from the solution to the vacuum can be estimated with recently developed DFT calculations. The accuracy required is to discuss the energy difference's order and judge whether the rate increase is rational or not.”

Reviewer 3 raises an important problem of accurate evaluation of the solvent effects on the energetics of the electronically excited states. This is a notoriously difficult process when significant charge rearrangement occurs upon excitation, as common computational approaches often use unrelaxed ground state density to compute the solvent response.^[R2] This issue is further complicated by the possible reordering of the excited states in the solvent medium due to the necessity of a partial geometry optimization in the excited state. To address Reviewer 4's comment, we have attempted to visualize the electrostatic interactions with the solvent by comparing the charge distribution in the ground and excited states, shown in Supplementary Fig. 63, and have discussed the outcomes in the Supplementary Information. Furthermore, we evaluated the literature to estimate the possible increase in the rate constants associated with solvent removal. Thus, we considered the activation barriers for merocyanine-to-spiropyran isomerization to estimate changes in the rate constants.^[R3] For instance, this barrier is approximately 5 kcal/mol higher for calculations performed in a dielectric continuum corresponding to acetone in comparison with calculations performed in the gas phase.^[R3] Such decrease in activation energy corresponds to a ~4000-fold increase in rate constant ($T = \text{const}$), and therefore, these reported calculations are in line with the magnitude of the detected rate enhancement for spiropyran derivatives in a solvent-free environment in our work.

“The following report compares the SP MC activation barrier in the solution and adsorbed on a solid surface. The authors should consider citing the article. “Mamun et al. Chemistry of the photoisomerization and thermal reset of nitro-spiropyran and merocyanine molecules on the channel of the MoS2 field effect transistor. Phys. Chem. Chem. Phys. 2021, 23, (48), 27273-27281””

We completely agree with Reviewer 3 about the mentioned studies, which are highly relevant to our work. These studies were included as reference 14, and we have highlighted this reference in the revised manuscript.

“The paragraph starting from 248 has a length of a page. Tough the authors describe in detail of the similarities and differences between molecules, the referee had a hard to time understanding the flow of the story. This is also true for the following two paragraphs, which start with the same phrase of ‘Similar to’. The authors should check whether the description can be more organized or not.”

We thank Reviewer 3 for mentioning how we could improve the clarity of our manuscript. We have restructured the mentioned paragraph in the revised version of the manuscript.

Reviewer 4

“The submitted manuscript (NCOMMS-23-33281) was reviewed and it could be accepted for publication in Nature Communications after considering the following major corrections.”

We appreciate Reviewer 4's detailed and constructive feedback which we have addressed below.

“At the beginning of "Introduction", Ref #1-11 has been cited twice. It is better to split them and specify each to the relevant subject.”

We completely agree with Reviewer 4's suggestion and have updated the revised manuscript to specify the references relevant to each subject.

“Page 2, Line 51-52: Please note that what you are talking about is “rate constant” and not “rate”.”

We thank Reviewer 4 for pointing out how we could be more accurate with our language and have changed “rate” to “rate constant” in the revised version of the manuscript.

“Scheme 1” has not been cited in the text. Also, the description for the right scheme in caption ((right) a solid solvent-free...) does not conform with the scheme (gas-phase analog).”

We appreciate Reviewer 4’s close reading of our manuscript. We have revised the caption of Scheme 1 and referenced it in the text.

“Page 5, Line 113-115: It is suggested to provide crystallographic data for diarylethens to confirm the statement about their minimal structural rearrangement.”

We completely agree with Reviewer 4. We have included X-ray crystal structures of a diarylethene derivative in the open and closed forms (Supplementary Fig. 62) to support our claim that its photoisomerization is associated with minimal structural rearrangement. We have also included a reference to this figure in the revised version of the manuscript.

“Page 7, Line 144-145: Please check that the hydrazones with pyridyl groups are 7-9 (Figure 1). Also, check Table 1 for this issue.”

We are grateful that Reviewer 4 has brought this potentially confusing point to our attention. Photoisomerization of hydrazone derivatives with pyridyl groups integrated as pillars within a MOF caused framework degradation. Instead, carboxylic acid-functionalized hydrazone derivatives (**5** and **6**) were integrated within UiO-67, the structural integrity of which was confirmed after hydrazone isomerization. We have revised the mentioned section of the manuscript to better explain these aspects.

“Table 1: What are the numbers in parenthesis in the “k” column, i.e. 0.033(5), 0.06(7) and etc.? Define them properly in the text or Table footnote.”

We appreciate Reviewer 4 for pointing out this oversight. The numbers given in parentheses refer to the standard deviation with respect to the last digit given for the average rate constant values (i.e., 0.033(5) corresponds to 0.033 ± 0.005). We have clarified this aspect in the Table footnote.

“Figure 4: For better elucidation and conformation with the text and Table 1, you may replace “evacuation” (in the caption) or “(evac)” (in the right spectrum) with “solvent-free”.”

We completely agree with Reviewer 4’s suggestion which will improve the consistency of our manuscript and have replaced “evac” with “solvent-free” in Figure 4. We have also clarified that the mentioned sample was solvent-free in the figure caption.

“Page 12, Line 255-260: The authors talk about steric factors for hydrazones as the controlling parameter on the kinetic of photoisomerization. Why didn’t they note to the dipole-dipole interactions for E and Z isomers during such photo-transformation? This seems to be important for hydrazones, while they would be ignored for diarylethens. However, this parameter and more have comprehensively been studied before for spiropyran (J. Mater. Chem. C, 2017, 5, 6588—6600) and the authors may refer to it.”

Reviewer 4 makes an excellent point that the photoisomerization of hydrazone derivatives can be affected by possible dipole-dipole interactions for the *E*- or *Z*-isomers. Solvent-dependent isomerization of hydrazone derivatives has been studied extensively by the Arahamian group (one of the co-authors), for example, who demonstrated that the pH-induced isomerization rate for hydrazone derivatives could be varied within one order of magnitude depending on solvent polarity.^[R4] To investigate this possibility using our system, we have measured the photoisomerization rate constants for compound **5**, which were selected for integration in MOFs, in more than one solvent. Like in other studies in literature, a change in photoisomerization rate constant within one order of magnitude was detected for **5** in ethanol versus DMF. However, since we did not detect drastic changes in photoisomerization rate constants for hydrazone derivatives in solvent-free environments, we hypothesize that the dominant factor which results in enhanced photoisomerization in the absence of solvent is the presence of zwitterionic species rather than the mentioned dipole-dipole interactions. In addition, we have referenced the suggested studies in the revised version of the manuscript as reference 63.

“In continuum to comment #8 and with respect to Table 1, the authors have not assessed the role of variation in solvent polarity on the photoisomerization process. It is recommended to give rate constants for one of the hydrazones in solvents with different polarities. This may improve their discussion and deduction about the interaction of E and Z isomers with the employed MOF groups.”

We completely agree with Reviewer 4’s suggestion and have measured the photoisomerization rate constant for hydrazone derivatives in ethanol, DMF, and toluene (Table 1). Similar to other literature reports involving hydrazone derivatives, we detected small enhancements in photoisomerization rate constant (within one order of magnitude) for hydrazone derivatives with decreasing polarity of the organic solvent.

“The authors need to explain the importance of drastic enhancement in the photoisomerization rate (constant) in “Introduction” and “Results and discussion” properly. What applications will be developed and weaknesses will be covered by this potentiality? Some few points have been mentioned in “Conclusion” very briefly, but I think it needs to be emphasized more.”

We thank Reviewer 4 for the suggestion which will definitely clarify the context of our work and highlight the potential future applications. We have modified the introduction to emphasize these aspects in the revised version of the manuscript.

REVIEWERS' COMMENTS

Reviewer #1 (Remarks to the Author):

The authors successfully addressed my points and adequately shared their views on my suggestion. I read other reviewers' comments and their responses too, and they furthered the readability of the manuscript. I am pleased to recommend publishing this manuscript as revised.

Reviewer #2 (Remarks to the Author):

All comments were adequately addressed, and the current version of manuscript can be published as is.

Reviewer #3 (Remarks to the Author):

The authors revised their manuscript adequately according to the reviewer's comments. The manuscript should be published as it is.

Reviewer #4 (Remarks to the Author):

After considering the requested revisions, the manuscript can be accepted for publication now.

Reviewer 1

“The authors successfully addressed my points and adequately shared their views on my suggestion. I read other reviewers' comments and their responses too, and they furthered the readability of the manuscript. I am pleased to recommend publishing this manuscript as revised.”

We thank Reviewer 1 for their careful reading of our revised manuscript, including the comments made by the other Reviewers.

Reviewer 2

“All comments were adequately addressed, and the current version of manuscript can be published as is.”

We appreciate Reviewer 2 for considering our responses to each of the points that they raised.

Reviewer 3

“The authors revised their manuscript adequately according to the reviewer's comments. The manuscript should be published as it is.”

We are grateful for Reviewer 3's suggestions, which improved our manuscript, and for their reading of the revised version.

Reviewer 4

“After considering the requested revisions, the manuscript can be accepted for publication now.”

Reviewer 4's comments were very constructive and improved the quality of our work. We are thankful for their reading of our revised manuscript.